# The association between variables of cardiopulmonary exercise test and quality of life in patients with chronic Chagas cardiomyopathy (Insights from the PEACH STUDY)

Marcelo Carvalho Vieira[1,2]*, Fernanda de Souza Nogueira Sardinha Mendes[1], Paula Simplício da Silva[1], Gilberto Marcelo Sperandio da Silva[1], Flavia Mazzoli-Rocha[1], Andrea Silvestre de Sousa[1], Roberto Magalhães Saraiva[1], Marcel de Souza Borges Quintana[1], Henrique Silveira Costa[3], Vitor Barreto Paravidino[4,5], Luiz Fernando Rodrigues, Junior[6,7], Alejandro Marcel Hasslocher-Moreno[1], Pedro Emmanuel Alvarenga Americano do Brasil[1], Mauro Felippe Felix Mediano[1,6]

1 Evandro Chagas National Institute of Infectious Disease, Oswaldo Cruz Foundation, Rio de Janeiro, Rio de Janeiro, Brazil, 2 Center for Cardiology and Exercise, Aloysio de Castro State Institute of Cardiology, Rio de Janeiro, Rio de Janeiro, Brazil, 3 Department of Physical Therapy, Federal University of Jequitinhonha and Mucuri Valleys, Diamantina, Minas Gerais, Brazil, 4 Institute of Social Medicine, State University of Rio de Janeiro, Rio de Janeiro, Rio de Janeiro, Brazil, 5 Department of Physical Education and Sports, Naval Academy, Rio de Janeiro, Rio de Janeiro, Brazil, 6 Department of Research and Education, National Institute of Cardiology, Rio de Janeiro, Rio de Janeiro, Brazil, 7 Department of Physiological Sciences, Federal University of the State of Rio de Janeiro, Rio de Janeiro, Rio de Janeiro, Brazil

* vieiramc@yahoo.com.br

## Abstract

Studies investigating the association between functional capacity and quality of life (QoL) in individuals with chronic Chagas cardiomyopathy (CCC) usually do not include a gold-standard evaluation of functional capacity, limiting the validity and the interpretation of the results. The present study is a cross-section analysis aiming to evaluate the association between functional capacity (quantified by cardiopulmonary exercise test [CPET]) and QoL in individuals with CCC. QoL was assessed using the SF-36 questionnaire. Sociodemographic, anthropometric, clinical, cardiac function and maximal progressive CPET variables were obtained from PEACH study. Generalized linear models adjusted for age, sex, and left ventricular ejection fraction were performed to evaluate the association between CPET variables and QoL. After adjustments, $VO_2$ peak and $VO_2$ AT were both associated with physical functioning ($\beta$ = +0.05 and $\beta$ = +0.05, respectively) and physical component summary ($\beta$ = +0.03 and $\beta$ = +0.03, respectively). Double product was associated with physical functioning ($\beta$ = +0.003), general health perceptions ($\beta$ = +0.003), physical component summary ($\beta$ = +0.002), and vitality ($\beta$ = +0.004). HRR≤12bpm was associated with physical functioning ($\beta$ = -0.32), role limitations due to physical problems ($\beta$ = -0.87), bodily pain ($\beta$ = -0.26), physical component summary ($\beta$ = -0.21), vitality ($\beta$ = -0.38), and mental health ($\beta$ = -0.19). $VE/VCO_2$ slope presented association with all mental scales of SF-36: vitality ($\beta$ = -0.028), social functioning ($\beta$ = -0.024), role limitations due to emotional problems ($\beta$ = -0.06), mental

**Data Availability Statement:** The data underlying the results presented in the study are available at osf.io/gkavb.

**Funding:** The author(s) received no specific funding for this work.

**Competing interests:** The authors have declared that no competing interests exist.

health (β = -0.04), and mental component summary (β = -0.02). The associations between CPET variables and QoL demonstrate the importance of CPET inclusion for a more comprehensive evaluation of individuals with CCC. In this setting, intervention strategies aiming to improve functional capacity may also promote additional benefits on QoL and should be incorporated as a treatment strategy for patients with CCC.

## Introduction

Chagas disease (CD) is a parasitic infection caused by the protozoan Trypanosoma cruzi [1] and considered a neglected disease by the World Health Organization [2]. Traditionally restricted to rural underdeveloped areas of Central and South America, the increase of migratory flow observed in last decades transformed CD into a health issue in several nonendemic countries [1,3].

During the chronic phase, approximately 20 to 40% of the CD infected individuals may develop the cardiac form of the disease, a condition usually known as chronic Chagas cardiomyopathy (CCC) [4,5]. CCC is characterized by a persistent inflammatory process and the development of myocardial fibrosis, leading to arrhythmias, thromboembolism, and heart failure (HF) [5,6] that negatively impact the quality of life (QoL).

Recently, improvements on QoL have become a therapeutic goal for the management of patients with several chronic diseases, with its evaluation gaining progressively importance [7,8], despite the lack of standardization of the results from longitudinal analysis which may compromise the comparison of results between trials and jeopardize their clinical applications [9]. QoL can predict death and hospitalization in individuals with HF [10], as well as adverse cardiovascular outcomes in CCC patients [11]. Individuals with CD present lower scores of QoL [12–19] and individuals with CCC present lower QoL when compared to healthy individuals [14], to those with the indeterminate form of CD [12,16], and to those with HF from other etiologies [15,20,21].

The association between functional capacity and QoL has been previously demonstrated in individuals with HF [22] and CCC [23–30]. However, studies investigating this association in individuals with CCC usually included submaximal evaluations and indirect measures of functional capacity, limiting the validity and the interpretation of the results obtained until now. Therefore, the study of the association between QoL and functional capacity measured by cardiopulmonary exercise test (CPET), the gold standard method that directly assesses functional capacity by gas exchange ratio [31], can provide a more accurate and precise information of individuals with CCC, allowing the development of tailored strategies to improve QoL in this population.

## Methods

### Study design

This is a secondary analysis using cross-sectional baseline data from PEACH study, a single center, superiority randomized parallel-group clinical trial of exercise training versus a control group with no exercise training, conducted from March 2015 to January 2017 at the Evandro Chagas National Institute of Infectious Diseases (INI) of Oswaldo Cruz Foundation (Fiocruz). The sample comprised 30 CD patients (confirmed by two distinct serological tests) of both sexes, older than 18 years, with CCC, left ventricular ejection fraction (LVEF) <45% or HF

symptoms (CCC stages B2 or C), New York Heart Association class I or II for at least three months, and clinically stable and under optimal medical therapy according to HF guidelines for at least six weeks. Exclusion criteria were motor or musculoskeletal limitations that preclude the exercise training, pregnancy, unavailability to attend exercise sessions 3 times a week, practice of regular exercise training (>1 week) in the three months prior to the study, smoking, or evidence of non-CCC cardiomyopathies. A complete description and the main results of the PEACH study have been previously published [32,33].

## Measurements

Sociodemographic and clinical variables were assessed during the initial assessment, together with a maximal progressive CPET, QoL questionnaire, anthropometric and cardiac function evaluations, which were performed within a one-week range.

Sociodemographic variables were obtained through interviews and included age, sex, income, schooling, and self-reported race. Income was stratified into two categories (<2 and ≥2 minimum wages per month). Schooling included the years of formal study, stratified into three categories (<5 years, 5–9 years, and >9 years). Clinical variables were obtained from medical records and included stage of CCC, presence of arterial hypertension, diabetes mellitus, dyslipidemia, history of stroke, presence of arrhythmias, cardiac devices, and medications.

QoL was assessed using the Medical Outcomes Study 36-Item Short-form of Health Survey (SF-36) questionnaire [34,35], translated into Portuguese and validated for the Brazilian population [36], by a single interviewer. SF-36 is a generic multidimensional instrument, composed of 36 questions, referring to the four-week period prior to the interview, and divided into eight different scales: physical functioning, role limitations due to physical problems, bodily pain, general health perceptions, vitality, social functioning, role limitations due to emotional problems and mental health. These scales define two summary scores: physical component summary (PCS) and mental component summary (MCS). Participants receive a final score ranging from zero (worst QoL) to 100 (best QoL) [34,35].

Maximal symptom-limited CPET was performed in a treadmill (Inbramed, Brazil) with a ramp protocol and active recovery, using a $VO_{2000}$ gas analyzer (MedGraphics, St. Paul, MN) connected to a computerized Ergo PC Elite system (Micromed, Brazil), with patients under use of their standard medications. The following CPET variables were assessed: oxygen consumption at peak of exercise ($VO_2$ peak), percent achieved of predicted oxygen uptake at peak of exercise ($\%PPVO_2$), oxygen consumption at anaerobic threshold ($VO_2$ AT), double product, minute ventilation-carbon dioxide production relationship ($VE/VCO_2$ slope), $O_2$ pulse, oxygen uptake efficiency slope (OUES), and heart rate recovery at the first minute (HRR). The $VO_2$ peak was defined as the highest value 30 seconds before or after the maximum effort or the plateau in oxygen uptake, and the anaerobic threshold (AT) by the V-slope method together with the ventilatory equivalents for $VO_2$ and carbon dioxide production ($VCO_2$), used to identify ventilatory thresholds [37]. The double product was calculated as a product of heart rate and systolic blood pressure at the peak of exercise. HRR was defined as the difference between maximal exercise heart rate and the heart rate at the first minute in the recovery phase, stratified into two categories (≤12 beats and >12 beats) [38]. The Ergo PC Elite software determined the other variables obtained on the CPET.

The anthropometric evaluation consisted of measurements of height, weight, and waist-to-hip ratio [39]. Body mass index (BMI) was calculated as the ratio of weight (kg) to height squared ($m^2$) and classified according to WHO definition [40].

Cardiac function was assessed by transthoracic echocardiogram following the American Society of Echocardiography recommendations, using a phased-array ultrasound system

(Vivid 7, GE Medical Systems, Milwaukee, WI) equipped with a M4S phased-array transducer [41]. LVEF was determined using the modified Simpson's rule.

## Data analysis

Descriptive analysis of sociodemographic, anthropometric, clinical, cardiac function and CPET variables consisted of mean and standard deviation for continuous variables and frequency and percentage for categorical variables. Descriptive analysis of QoL scores consisted of mean, standard deviation and range. The association between CPET variables (exposure variables) and QoL scales (outcomes) was determined by generalized linear models with gamma distribution and log-link function that accounts for skewed and heteroscedastic residuals distribution. Regression models were performed without adjustments and adjusted for age, sex, and LVEF that were considered as potential confounders according to the literature [5]. The partial eta-squared (partial η2), the proportion of variance in the dependent variable explained by each term in the model, were determined for each CPET variable in separate unadjusted and adjusted models.

The Research Electronic Data Capture (REDCap) web application was used for data management and the data analysis was conducted using R software (version 3.6.2). An association matrix graph was built to visually demonstrate the eta-squared for the association between CPET variables and scales and summary scores of SF-36 using the command ggplot2 in R Studio software. Statistical significance was set at $p \leq 0.05$ for all analyses.

## Ethical considerations

All participants read and signed a written informed consent, and received information about the goals and procedures of the study. The study was performed in accordance with the resolution 466/2012 of the Brazilian National Council of Health and was approved by the Evandro Chagas National Institute of Infectious Diseases Research Ethics Committee (CAAE: 38038914.6.0000.5262; report number 3.165.034) in February 27th, 2015. The clinical trial was registered at ClinicalTrials.gov (NCT02517632).

## Results

The characteristics of the patients included in the study are shown in Table 1. Briefly, the mean age was 59.8 ± 10.0 years, with 66.7% males, 60.0% non-white, 56.7% in the low-income group, and 86.7% had up to nine years of schooling. The majority (73.3%) was classified as stage C of CCC (with HF).

The prevalence of hypertension was 3.3%, diabetes mellitus was 16.7%, dyslipidemia was 30.0%, and history of stroke was 16.7%. Most participants (73.3%) presented arrhythmia and 46.7% used a cardiac device. The mean LVEF was 33.1% (± 7.8). Regarding medications in use, 93.3% of the participants were treated with beta-blockers, 93.3% with angiotensin-converting enzyme inhibitors or angiotensin receptor blockers, 50% with aldosterone antagonist, and 73.3% were taking diuretics. For the variables from CPET, the mean $VO_2$ peak was 16.5 (± 5.6) ml.kg$^{-1}$.min$^{-1}$, $VO_2$ AT was 14.8 (± 4.2) ml.kg$^{-1}$.min$^{-1}$, and VE/VCO$_2$ slope was 29.3 (± 6.3). Sixteen individuals (53.3%) presented HRR equal to or lesser than 12 bpm (Table 1).

The description of QoL scores by each scale is depicted in Table 2. Overall, patients presented lower scores for the scales related to physical aspects in comparison to those related to mental aspects. The role limitations due to physical problems (60.0 ± 45.3) and general health perceptions (62.2 ± 22.2) presented the lowest scores, while social functioning (83.1 ± 23.6) and mental health (81.2 ± 20.2) achieved the highest scores. The summary scores were 43.0 (± 9.8) for PCS and 53.0 (± 11.7) for MCS.

**Table 1. Characteristics of participants included in the study (n = 30).**

| Variable | Frequency (percentage) or Mean ±standard deviation |
|---|---|
| *Sociodemographic variables* | |
| Age (years) | 59.8 ±10.0 |
| Sex (%) | |
| Female | 10 (33.3) |
| Male | 20 (66.7) |
| Income (%) | |
| < 2 minimum wage | 17 (56.7) |
| ≥ 2 minimum wage | 13 (43.3) |
| Schooling (%) | |
| <5 years | 13 (43.3) |
| 5–9 years | 13 (43.3) |
| >9 years | 4 (13.3) |
| Race (%) | |
| White | 12 (40.0) |
| Mulatto | 14 (46.7) |
| Black | 3 (10.0) |
| Indigenous | 1 (3.3) |
| *Clinical variables* | |
| Clinical form of CCC (%) | |
| B2 (without heart failure) | 8 (26.7) |
| C (with heart failure) | 22 (73.3) |
| Hypertension (%) | 1 (3.3) |
| Diabetes Mellitus (%) | 5 (16.7) |
| Dyslipidemia (%) | 8 (30.0) |
| Previous stroke (%) | 5 (16.7) |
| Arrhythmia (%) | 22 (73.3) |
| Cardiac device (%) | 14 (46.7) |
| LVEF (%) | 33.1 ±7.8 |
| Medications (%)[†] | |
| Beta-blocker | 28 (93.3) |
| Diuretics | 22 (73.3) |
| Angiotensin-converting enzyme inhibitors | 16 (53.3) |
| Aldosterone antagonist | 15 (50.0) |
| Anticoagulants | 14 (46.7) |
| Angiotensin receptor blockers | 12 (40.0) |
| Digital | 7 (23.3) |
| Amiodarone | 6 (20.0) |
| *Anthropometric variables* | |
| Weight (Kg) | 66.5 ±14.0 |
| Height (m) | 1.60 ±0.1 |
| BMI (Kg/m$^2$) | 25.4 ±5.2 |
| BMI classification (%) | |
| Underweight | 2 (6.7) |
| Eutrophic | 14 (46.7) |
| Overweight | 9 (30.0) |
| Obese | 5 (16.7) |
| Waist-to-hip ratio | |

(*Continued*)

**Table 1.** (Continued)

| Variable | Frequency (percentage) or Mean ±standard deviation |
|---|---|
| Female | 0.88 ±0.09 |
| Male | 0.92 ±0.07 |
| *CPET variables* | |
| VO$_2$ peak (ml.kg$^{-1}$.min$^{-1}$) | 16.5 ±5.6 |
| %PPVO$_2$ | 55.5 ±15.6 |
| VO$_2$ AT (ml.kg$^{-1}$.min$^{-1}$)$^¥$ | 14.8 ±4.2 |
| Double product (mmHg.bpm) x10$^2$ | 135.5 ±50.7 |
| O$_2$ Pulse (L/sys) | 10.1 ±3.6 |
| VE/VCO$_2$ slope | 29.3 ±6.3 |
| OUES | 1.4 ±0.7 |
| HRR $\leq$ 12 bpm (%) | 53.3 (16) |

CCC: Chronic Chagas cardiomyopathy; LVEF: Left ventricular ejection fraction; BMI: Body mass index; CPET: Cardiopulmonary exercise test; VO$_2$ peak: Oxygen consumption at peak exercise; %PPVO$_2$: Percent achieved of predicted oxygen uptake at peak exercise; VO$_2$ AT: Oxygen consumption at anaerobic threshold; VE/VCO2 slope: Minute ventilation-carbon dioxide production relationship; OUES: Oxygen uptake efficiency slope; HRR: First-minute heart rate recovery.

$^¥$ VO$_2$ AT: n = 17.

[†] Medications: Beta-blocker: Carvedilol; Diuretics: Furosemide and hydrochlorothiazide; Angiotensin-converting enzyme inhibitors: Enalapril and captopril; Aldosterone antagonist: Spironolactone; Anticoagulants: Warfarin; Angiotensin receptor blockers: Losartan; Digital: Digoxin.

Table 3 presents the association between CPET variables and physical scales of QoL. After adjustments for potential confounders, VO$_2$ peak and VO$_2$ AT were both positively associated with physical functioning (β = +0.05 95%CI +0.03 to +0.07 and β = +0.05 95%CI +0.02 to +0.08, respectively) and PCS (β = +0.03 95%CI +0.01 to +0.05 and β = +0.03 95%CI +0.01 to +0.06, respectively). Double product was positively associated with physical functioning (β = +0.003 95%CI +0.000 to +0.007), general health perceptions (β = +0.003 95%CI +0.000 to +0.006), and PCS (β = +0.002 95%CI +0.000 to +0.004), whilst HRR $\leq$ 12 bpm was negatively associated with physical functioning (β = -0.32 95%CI -0.61 to -0.04), role limitations due to physical problems (β = -0.87 95%CI -1.53 to -0.21), bodily pain (β = -0.26 95%CI -0.49 to -0.04), and PCS (β = -0.21 95%CI -0.38 to -0.05). The CPET variables that most explained the QoL variation in the adjusted models were VO$_2$ AT (50% for physical functioning and 36% for PCS) and VO$_2$ peak (31% for physical functioning and 21% for PCS).

The association between CPET variables and mental scales of QoL is presented in Table 4. After adjustments for potential confounders, VE/VCO$_2$ slope presented a negative association with all mental scales of SF-36: vitality (β = -0.028 95%CI -0.055 to -0.002), social functioning (β = -0.024 95%CI -0.044 to -0.003), role limitations due to emotional problems (β = -0.06 95% CI -0.12 to +0.01), mental health (β = -0.04 95%CI -0.06 to -0.02), and MCS (β = -0.02 95%CI -0.04 to -0.01). HRR $\leq$ 12 bpm was negatively associated with vitality (β = -0.38 95%CI -0.68 to -0.08) and mental health (β = -0.19 95%CI -0.38 to -0.01). Double product was positively associated with vitality (β = +0.004 95%CI 0.000 to +0.007). The CPET variables that most explained the QoL variation in the adjusted models were VE/VCO$_2$ slope (45% for mental health and 31% for MCS) and HRR $\leq$ 12 bpm (20% for vitality).

Fig 1 illustrates the eta-squared for the association between CPET variables and scales and summary scores of SF-36.

**Table 2. Quality of life assessed by SF-36 (n = 30).**

| Variable | Mean ±standard deviation | Range |
|---|---|---|
| SF-36 QoL Scales | | |
| Physical functioning | 65.8 ±26.5 | 15–100 |
| Role limitations due to physical problems | 60.0 ±45.3 | 0–100 |
| Bodily pain | 73.5 ±23.6 | 31–100 |
| General health perceptions | 62.2 ±22.2 | 25–100 |
| Physical Component Summary | 43.0 ±9.8 | 24–59 |
| Vitality | 66.8 ±26.8 | 5–100 |
| Social functioning | 83.1 ±23.6 | 13–100 |
| Role limitations due to emotional problems | 65.6 ±43.3 | 0–100 |
| Mental health | 81.2 ±20.2 | 12–100 |
| Mental Component Summary | 53.0 ±11.7 | 19–66 |

SF-36: Medical Outcomes Study 36-Item Short-form of Health Survey; QoL: Quality of life.

## Discussion

The present study demonstrated a significant association between several CPET variables and QoL in patients with CCC. The variables that most explained the variation in the physical scales of QoL were $VO_2$ AT and $VO_2$ peak, whilst the variables that most explained the variation in the mental scales of QoL were VE/VCO$_2$ slope and HRR $\leq$ 12 bpm. Overall, physical scales presented lower scores in comparison to mental scales of QoL, which may reflect the impaired functional capacity observed in individuals with CCC [42].

The use of normative QoL values can provide important insights about the QoL status of a specific group, allowing the comparison with other populations [43]. In this sense, comparing the SF-36 results from our sample with the Brazilian normative data [44], CCC patients presented lower QoL values than the general population for physical functioning, role limitation to physical problems, general health perceptions, and role limitation to emotional problems scales, as well as in the PCS. However, previous studies showed different results for the physical functioning scale [11,14,15,27,45,46], with mean scores ranging from 18.8 [45] to 85.0 [14,27]. The differences observed across studies may be explained by the heterogeneity of the studied groups in terms of clinical characteristics, cardiac function, functional class, and geographical origin.

Regarding the association between CPET variables and QoL, some interesting findings were observed. VO2 peak, %PPVO$_2$ and VO2 AT correspond to maximal and submaximal parameters of functional capacity, respectively [31]. In the present study, all these three parameters were positively associated with physical functioning and PCS. Similarly, other studies identified an association between QoL and functional capacity in individuals with CCC [23–30], although most of them used field tests [23,24,28,29] or questionnaires [30] to estimate the functional capacity. In line with our results, Andersen et al. (2018) found a positive correlation between VO2 AT and PCS in patients with cardiac disease [47]. On the other hand, Ritt et al. (2013) and Costa et al. (2014), found a significant correlation between VO2 peak and QoL measured, respectively, by Minnesota Living with Heart Failure Questionnaire (MLHFQ) and SF-36 in patients with CCC [26,27]. In addition, Ritt et al. (2012) identified a significant difference in the QoL between the groups with VO2 peak > and $\leq$ 12 ml.kg-1.min-1 (a threshold for indication of heart transplantation), with greater QoL scores being observed among those individuals in the greater VO2 peak group [25].

**Table 3. Association between CPET variables and QoL physical related scales.**

| CPET variables | SF-36 Physical functioning domain | | | |
|---|---|---|---|---|
| | Unadjusted | | Adjusted[§] | |
| | β (95% CI) | eta-squared | β (95% CI) | eta-squared |
| $VO_2$ peak (ml.$kg^{-1}$.$min^{-1}$) | **+0.05 (+0.03 to +0.07)** | 0.38 | **+0.05 (+0.02 to +0.09)** | 0.31 |
| %$PPVO_2$ | **+0.02 (+0.01 to +0.03)** | 0.42 | **+0.02 (+0.01 to +0.03)** | 0.33 |
| $VO_2$ AT (ml.$kg^{-1}$.$min^{-1}$)[¥] | **+0.04 (+0.02 to +0.06)** | 0.48 | **+0.05 (+0.02 to +0.08)** | 0.50 |
| Double product ($x10^{-2}$) | **+0.004 (+0.001 to +0.007)** | 0.21 | **+0.003 (+0.000 to +0.007)** | 0.15 |
| $O_2$ Pulse (ml/sys) | +0.036 (-0.005 to +0.076) | 0.10 | +0.022 (-0.026 to +0.069) | 0.03 |
| VE/$VCO_2$ slope | **-0.024 (-0.047 to -0.001)** | 0.13 | -0.018 (-0.044 to +0.009) | 0.06 |
| OUES ($x10^{-3}$) | **+0.29 (+0.07 to +0.50)** | 0.19 | +0.24 (-0.06 to +0.54) | 0.09 |
| HRR $\leq$ 12 bpm (%) | -0.23 (-0.52 to +0.06) | 0.08 | **-0.32 (-0.61 to -0.04)** | 0.16 |
| | SF-36 Role limitations due to physical problems scale | | | |
| | Unadjusted | | Adjusted[§] | |
| | β (95% CI) | eta-squared | β (95% CI) | eta-squared |
| $VO_2$ peak (ml.$kg^{-1}$.$min^{-1}$) | +0.05 (-0.01 to +0.10) | 0.09 | +0.06 (-0.02 to +0.15) | 0.09 |
| %$PPVO_2$ | **+0.021 (+0.001 to +0.041)** | 0.13 | +0.023 (-0.004 to +0.050) | 0.10 |
| $VO_2$ AT (ml.$kg^{-1}$.$min^{-1}$)[¥] | +0.055 (-0.003 to +0.114) | 0.19 | +0.065 (-0.017 to +0.146) | 0.17 |
| Double product ($x10^{-2}$) | +0.006 (0.000 to +0.012) | 0.11 | +0.006 (-0.001 to +0.013) | 0.10 |
| $O_2$ Pulse (ml/sys) | +0.04 (-0.04 to +0.12) | 0.04 | +0.03 (-0.06 to +0.13) | 0.02 |
| VE/$VCO_2$ slope | -0.02 (-0.07 to +0.02) | 0.03 | -0.02 (-0.07 to +0.03) | 0.02 |
| OUES ($x10^{-3}$) | +0.41 (-0.04 to +0.86) | 0.10 | +0.46 (-0.18 to +1.09) | 0.07 |
| HRR $\leq$ 12 bpm (%) | -0.52 (-1.11 to +0.08) | 0.09 | **-0.87 (-1.53 to -0.21)** | 0.21 |
| | SF-36 Bodily pain scale | | | |
| | Unadjusted | | Adjusted[§] | |
| | β (95% CI) | eta-squared | β (95% CI) | eta-squared |
| $VO_2$ peak (ml.$kg^{-1}$.$min^{-1}$) | +0.020 (-0.001 to +0.041) | 0.11 | +0.01 (-0.02 to +0.04) | 0.02 |
| %$PPVO_2$ | +0.005 (-0.003 to +0.012) | 0.05 | +0.004 (-0.005 to +0.014) | 0.03 |
| $VO_2$ AT (ml.$kg^{-1}$.$min^{-1}$)[¥] | +0.02 (-0.01 to +0.05) | 0.14 | +0.01 (-0.03 to +0.05) | 0.02 |
| Double product ($x10^{-2}$) | +0.002 (-0.001 to +0.004) | 0.07 | +0.002 (0.000 to +0.005) | 0.13 |
| $O_2$ Pulse (ml/sys) | 0.00 (-0.03 to +0.03) | 0.00 | -0.01 (-0.05 to +0.03) | 0.01 |
| VE/$VCO_2$ slope | -0.017 (-0.035 to +0.002) | 0.10 | -0.014 (-0.035 to +0.007) | 0.06 |
| OUES ($x10^{-3}$) | +0.07 (-0.11 to +0.25) | 0.02 | +0.04 (-0.19 to +0.28) | 0.01 |
| HRR $\leq$ 12 bpm (%) | -0.14 (-0.38 to +0.09) | 0.05 | **-0.26 (-0.49 to -0.04)** | 0.18 |
| | SF-36 General health perceptions scale | | | |
| | Unadjusted | | Adjusted[§] | |
| | β (95% CI) | eta-squared | β (95% CI) | eta-squared |
| $VO_2$ peak (ml.$kg^{-1}$.$min^{-1}$) | +0.01 (-0.01 to +0.03) | 0.02 | +0.02 (-0.01 to +0.06) | 0.07 |
| %$PPVO_2$ | +0.005 (-0.003 to +0.013) | 0.05 | +0.007 (-0.004 to +0.018) | 0.05 |
| $VO_2$ AT (ml.$kg^{-1}$.$min^{-1}$)[¥] | +0.03 (-0.01 to +0.06) | 0.11 | +0.047 (-0.004 to +0.098) | 0.22 |
| Double product ($x10^{-2}$) | **+0.003 (0.000 to +0.005)** | 0.15 | **+0.003 (0.000 to +0.006)** | 0.16 |
| $O_2$ Pulse (ml/sys) | +0.02 (-0.01 to +0.06) | 0.06 | +0.03 (-0.01 to +0.07) | 0.07 |
| VE/$VCO_2$ slope | -0.01 (-0.03 to +0.01) | 0.05 | -0.02 (-0.04 to +0.01) | 0.06 |
| OUES ($x10^{-3}$) | +0.15 (-0.04 to +0.35) | 0.08 | +0.26 (-0.02 to +0.53) | 0.12 |
| HRR $\leq$ 12 bpm (%) | **-0.27 (-0.52 to -0.01)** | 0.13 | -0.27 (-0.54 to +0.01) | 0.12 |
| | SF-36 Physical Component Summary | | | |
| | Unadjusted | | Adjusted[§] | |
| | β (95% CI) | eta-squared | β (95% CI) | eta-squared |
| $VO_2$ peak (ml.$kg^{-1}$.$min^{-1}$) | **+0.02 (+0.01 to +0.03)** | 0.21 | **+0.03 (+0.01 to +0.05)** | 0.21 |

*(Continued)*

**Table 3.** (Continued)

| CPET variables | SF-36 Physical functioning domain | | | |
| --- | --- | --- | --- | --- |
| | Unadjusted | | Adjusted[§] | |
| | β (95% CI) | eta-squared | β (95% CI) | eta-squared |
| %PPVO2 | **+0.007 (+0.002 to +0.012)** | 0.23 | **+0.008 (+0.002 to +0.015)** | 0.21 |
| $VO_2$ AT (ml.kg$^{-1}$.min$^{-1}$)[¥] | **+0.023 (+0.005 to +0.041)** | 0.29 | **+0.032 (+0.008 to +0.057)** | 0.36 |
| Double product (x10$^{-2}$) | **+0.002 (+0.001 to +0.004)** | 0.22 | **+0.002 (0.000 to +0.004)** | 0.20 |
| $O_2$ Pulse (ml/sys) | +0.02 (-0.01 to +0.04) | 0.06 | +0.01 (-0.02 to +0.04) | 0.03 |
| $VE/VCO_2$ slope | -0.01 (-0.02 to +0.01) | 0.04 | -0.01 (-0.02 to +0.01) | 0.02 |
| OUES (x10$^{-3}$) | **+0.14 (+0.01 to +0.26)** | 0.14 | +0.15 (-0.02 to +0.32) | 0.11 |
| HRR $\leq$ 12 bpm (%) | **-0.18 (-0.34 to -0.02)** | 0.15 | **-0.21 (-0.38 to -0.05)** | 0.21 |

Estimates in **bold** are statistically significant.

[§]Model adjusted for age, sex, and left ventricular ejection fraction.

[¥]$VO_2$ AT: n = 17.

CPET: Cardiopulmonary exercise test; QoL: Quality of Life; SF-36: Medical Outcomes Study 36-Item Short-form of Health Survey; $VO_2$ peak: Oxygen consumption at peak exercise; %PPVO$_2$: Percent achieved of predicted oxygen uptake at peak exercise; $VO_2$ AT: Oxygen consumption at anaerobic threshold; $VE/VCO_2$ slope: Minute ventilation-carbon dioxide production relationship; OUES: Oxygen uptake efficiency slope; HRR: First-minute heart rate recovery.

Other CPET variables were also associated with some QoL scales in our study. $VE/VCO_2$ slope and OUES represent the ventilatory efficiency [48], and its association with prognosis in HF [49,50] and in CCC [26] has already been demonstrated. Nogueira et al. (2010) examined 46 HF patients (28.3% with CCC) and found a significant negative correlation between peak $VE/VCO_2$ slope and role limitations due to physical problems scale of SF-36 [22]. In contrast, Arena et al. (2002) and Ritt et al. (2013) did not find any association between $VE/VCO_2$ slope and OUES with QoL measured through the MLHFQ in patients with HF [26,51]. Likewise, in the present study, neither $VE/VCO_2$ slope nor OUES showed any significant association with the physical scales of QoL. The unexpected poor correlation between $VE/VCO_2$ slope and physical scales of QoL can be explained by the high percentage (77%) of patients with normal $VE/VCO_2$ slope levels (<32.5) in the studied population [26], with a low impact on the ability to perform the activities that are evaluated in the SF-36 instrument. On the other hand, $VE/VCO_2$ slope (but not OUES) was inversely associated with all mental components of SF-36. Considering that most patients presented normal $VE/VCO_2$ slope levels, we can speculate that variations in normal levels of $VE/VCO_2$ slope may have impacted the performance of submaximal activities that required efforts greater than those activities evaluated in the physical scales of SF-36, negatively impacting the emotional aspects of QoL by the inability to perform these more physically demanding activities on daily living.

Autonomic dysfunction has been demonstrated in CCC patients [52,53] and may be identified by a blunted HRR after the peak exercise [54]. Low HRR after exercise tests has been demonstrated as evidence of poor prognosis and greater disease severity in patients with HF [55,56], even in submaximal tests [57], and may indicate the presence of autonomic dysfunction [38]. In the present study, HRR $\leq$ 12 bpm was inversely associated with both physical (physical functioning, role limitations due to physical problems, pain, and PCS) and mental (vitality and mental health) scales of SF-36. The possible mechanism to explain this finding may be the better autonomic regulation allowing a more adequate adjustment of heart rate and peripheral blood flow, which may result in optimization of peripheral energy consumption and reduction of the sensation of dyspnea and fatigue [58]. To our knowledge, there is no previous evidence about the relationship between autonomic modulation and QoL in HF

**Table 4. Association between CPET variables and QoL mental related scales.**

| CPET variables | SF-36 Vitality scale | | | |
|---|---|---|---|---|
| | Unadjusted | | Adjusted[§] | |
| | β (95% CI) | eta-squared | β (95% CI) | eta-squared |
| VO$_2$ peak (ml.kg$^{-1}$.min$^{-1}$) | **+0.03 (+0.003 to +0.055)** | 0.14 | +0.03 (-0.01 to +0.07) | 0.09 |
| %PPVO$_2$ | **+0.012 (+0.002 to +0.021)** | 0.17 | +0.010 (-0.002 to +0.022) | 0.09 |
| VO$_2$ AT (ml.kg$^{-1}$.min$^{-1}$)[¥] | +0.02 (-0.01 to +0.05) | 0.08 | +0.02 (-0.03 to +0.06) | 0.05 |
| Double product (x10$^{-2}$) | **+0.004 (+0.001 to +0.007)** | 0.18 | **+0.004 (0.000 to +0.007)** | 0.15 |
| O$_2$ Pulse (ml/sys) | +0.02 (-0.02 to +0.06) | 0.04 | +0.02 (-0.03 to +0.07) | 0.02 |
| VE/VCO$_2$ slope | **-0.029 (-0.052 to -0.006)** | 0.18 | **-0.028 (-0.055 to -0.002)** | 0.15 |
| OUES (x10$^{-3}$) | +0.19 (-0.03 to +0.42) | 0.09 | +0.18 (-0.13 to +0.49) | 0.05 |
| HRR ≤ 12 bpm (%) | -0.28 (-0.58 to +0.02) | 0.11 | **-0.38 (-0.68 to -0.08)** | 0.20 |
| | SF-36 Social functioning scale | | | |
| | Unadjusted | | Adjusted[§] | |
| | β (95% CI) | eta-squared | β (95% CI) | eta-squared |
| VO$_2$ peak (ml.kg$^{-1}$.min$^{-1}$) | +0.016 (-0.003 to +0.035) | 0.09 | +0.010 (-0.018 to +0.038) | 0.02 |
| %PPVO$_2$ | +0.005 (-0.002 to +0.012) | 0.06 | +0.004 (-0.005 to +0.013) | 0.03 |
| VO$_2$ AT (ml.kg$^{-1}$.min$^{-1}$)[¥] | +0.02 (-0.01 to +0.05) | 0.09 | +0.01 (-0.04 to +0.05) | 0.01 |
| Double product (x10$^{-2}$) | +0.002 (0.000 to +0.004) | 0.08 | +0.002 (-0.001 to +0.004) | 0.09 |
| O$_2$ Pulse (ml/sys) | 0.00 (-0.03 to +0.03) | 0.00 | -0.02 (-0.05 to +0.02) | 0.03 |
| VE/VCO$_2$ slope | **-0.024 (-0.040 to -0.007)** | 0.21 | **-0.024 (-0.044 to -0.003)** | 0.17 |
| OUES (x10$^{-3}$) | +0.09 (-0.06 to +0.25) | 0.05 | +0.06 (-0.17 to +0.29) | 0.01 |
| HRR ≤ 12 bpm (%) | -0.14 (-0.35 to +0.06) | 0.06 | -0.22 (-0.45 to +0.01) | 0.12 |
| | SF-36 Role limitations due to emotional problems scale | | | |
| | Unadjusted | | Adjusted[§] | |
| | β (95% CI) | eta-squared | β (95% CI) | eta-squared |
| VO$_2$ peak (ml.kg$^{-1}$.min$^{-1}$) | +0.041 (-0.005 to +0.087) | 0.10 | +0.030 (-0.042 to +0.102) | 0.03 |
| %PPVO$_2$ | +0.016 (-0.001 to +0.033) | 0.11 | +0.011 (-0.012 to +0.035) | 0.03 |
| VO$_2$ AT (ml.kg$^{-1}$.min$^{-1}$)[¥] | +0.04 (-0.01 to +0.09) | 0.13 | +0.01 (-0.08 to +0.09) | 0.00 |
| Double product (x10$^{-2}$) | +0.005 (-0.001 to +0.010) | 0.10 | +0.007 (0.000 to +0.014) | 0.13 |
| O$_2$ Pulse (ml/sys) | +0.04 (-0.03 to +0.11) | 0.05 | +0.03 (-0.06 to +0.12) | 0.02 |
| VE/VCO$_2$ slope | **-0.05 (-0.10 to -0.01)** | 0.18 | **-0.06 (-0.12 to -0.01)** | 0.16 |
| OUES (x10$^{-3}$) | +0.29 (-0.08 to +0.67) | 0.08 | +0.32 (-0.27 to +0.91) | 0.04 |
| HRR ≤ 12 bpm (%) | -0.16 (-0.64 to +0.31) | 0.02 | -0.36 (-0.96 to +0.24) | 0.05 |
| | SF-36 Mental health scale | | | |
| | Unadjusted | | Adjusted[§] | |
| | β (95% CI) | eta-squared | β (95% CI) | eta-squared |
| VO$_2$ peak (ml.kg$^{-1}$.min$^{-1}$) | +0.013 (-0.003 to +0.030) | 0.08 | +0.006 (-0.017 to +0.029) | 0.01 |
| %PPVO$_2$ | +0.006 (0.000 to +0.012) | 0.12 | +0.003 (-0.005 to +0.010) | 0.02 |
| VO$_2$ AT (ml.kg$^{-1}$.min$^{-1}$)[¥] | +0.012 (-0.021 to +0.045) | 0.03 | -0.002 (-0.047 to +0.042) | 0.00 |
| Double product (x10$^{-2}$) | +0.001 (0.000 to +0.003) | 0.07 | +0.001 (-0.001 to +0.003) | 0.05 |
| O$_2$ Pulse (ml/sys) | +0.01 (-0.01 to +0.04) | 0.02 | +0.01 (-0.02 to +0.04) | 0.01 |
| VE/VCO$_2$ slope | **-0.04 (-0.05 to -0.02)** | 0.44 | **-0.04 (-0.06 to -0.02)** | 0.45 |
| OUES (x10$^{-3}$) | +0.10 (-0.04 to +0.24) | 0.07 | +0.08 (-0.11 to +0.27) | 0.03 |
| HRR ≤ 12 bpm (%) | -0.12 (-0.31 to +0.06) | 0.06 | **-0.19 (-0.38 to -0.01)** | 0.15 |
| | SF-36 Mental Component Summary | | | |
| | Unadjusted | | Adjusted[§] | |
| | β (95% CI) | eta-squared | β (95% CI) | eta-squared |
| VO$_2$ peak (ml.kg$^{-1}$.min$^{-1}$) | +0.010 (-0.004 to +0.025) | 0.06 | +0.005 (-0.016 to +0.025) | 0.01 |

*(Continued)*

**Table 4.** (Continued)

| CPET variables | SF-36 Vitality scale | | | |
| --- | --- | --- | --- | --- |
| | Unadjusted | | Adjusted[§] | |
| | β (95% CI) | eta-squared | β (95% CI) | eta-squared |
| %PPVO$_2$ | +0.004 (-0.001 to +0.010) | 0.09 | +0.002 (-0.005 to +0.009) | 0.01 |
| VO$_2$ AT (ml.kg$^{-1}$.min$^{-1}$)[¥] | +0.01 (-0.02 to +0.04) | 0.03 | -0.01 (-0.04 to +0.03) | 0.01 |
| Double product (x10$^{-2}$) | +0.001 (0.000 to + 0.003) | 0.09 | +0.001 (0.000 to +0.003) | 0.09 |
| O$_2$ Pulse (ml/sys) | +0.008 (-0.015 to +0.030) | 0.02 | +0.004 (-0.023 to +0.030) | 0.00 |
| VE/VCO$_2$ slope | **-0.02 (-0.03 to -0.01)** | 0.31 | **-0.02 (-0.04 to -0.01)** | 0.31 |
| OUES (x10$^{-3}$) | +0.08 (-0.04 to +0.20) | 0.06 | +0.07 (-0.10 to +0.24) | 0.03 |
| HRR ≤ 12 bpm (%) | -0.08 (-0.24 to +0.08) | 0.04 | -0.14 (-0.30 to +0.03) | 0.10 |

Estimates in **bold** are statistically significant.

[§]Model adjusted for age, sex, and left ventricular ejection fraction.

[¥]VO$_2$ AT: n = 17.

CPET: Cardiopulmonary exercise test; QoL: Quality of Life; SF-36: Medical Outcomes Study 36-Item Short-form of Health Survey; VO$_2$ peak: Oxygen consumption at peak exercise; %PPVO$_2$: Percent achieved of predicted oxygen uptake at peak exercise; VO$_2$ AT: Oxygen consumption at anaerobic threshold; VE/VCO$_2$ slope: Minute ventilation-carbon dioxide production relationship; OUES: Oxygen uptake efficiency slope; HRR: First-minute heart rate recovery.

patients, despite some studies have shown that treatments aimed at improving autonomic regulation promoted an increase in QoL [59–62]. Nevertheless, our results are in line with those from van den Berg et al. (2001), that found an association between several autonomic function variables (deep breathing, isometric handgrip, standing up, head up tilting, and baroreflex sensitivity) and physical functioning, general health perceptions, vitality, and role limitations due

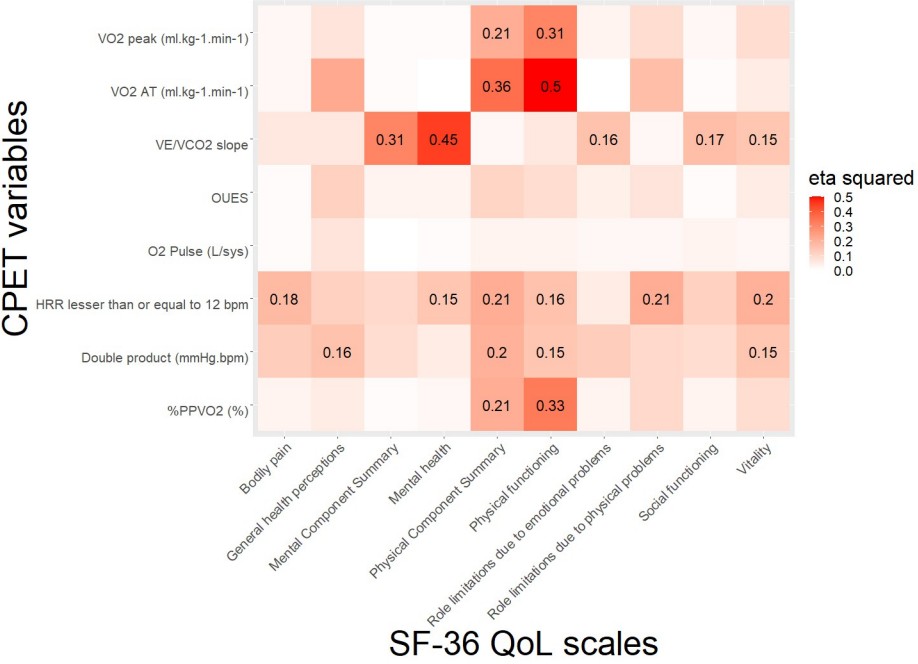

**Fig 1. Association matrix between CPET variables and quality of life (SF-36).** Shades of red indicate an increasing positive correlation coefficient. Only the significant correlation coefficient was shown.

to emotional problems scales of SF-36 in a sample of patients presenting paroxysmal atrial fibrillation [63].

In the present study, the double product was positively associated with physical functioning, general health perceptions, PCS, and vitality. Since the correlation between double product and $VO_2$ peak has already been demonstrated [64], we speculate that this association, especially on scales related to physical aspects, occurred because the higher value of the double product may express a higher functional capacity.

The major strength of our study was the inclusion of the CPET, the gold standard measure of functional capacity, which may allow for a more accurate assessment of the association between functional capacity and QoL in CCC patients. However, the small sample size was a limitation, with an a posteriori analysis demonstrating statistical power ranging from 5% to 97% (S1 Table). Moreover, our sample consisted of patients from an urban cohort and regularly followed at the outpatient clinic of a national reference center for the treatment of infectious disease, which may limit the applicability of the results for other populations. Besides that, the high-quality health care provided during the follow-up at a referral center may have positively affected the patients' perception of QoL. Although the use of beta-blockers does not appear to alter exercise capacity in maximal and submaximal tests, there appears to be a favorable effect on the VE/VCO2 slope [65]. Thus, as most of our sample was using beta-blockers (93.3%), this may have influenced the CPET response. Overweight and obesity are other variables that may influence the exercise test results [66] and were not included in the statistical model as potential confounders. Finally, depression may be an important confounding factor for QoL in individuals with CD [13,67] and was not assessed in the present study.

## Conclusions

The associations between CPET variables and QoL, especially for $VO_2$ AT and $VO_2$ peak with the physical scales, and VE/VCO$_2$ slope and HRR $\leq$ 12 bpm with the mental scales, reinforce the importance of CPET inclusion for a more comprehensive evaluation of individuals with CCC, when available. In this setting, intervention strategies aiming to improve functional capacity may also promote additional benefits on QoL and should be incorporated as a treatment strategy for patients with CCC.

## Supporting information

**S1 Table. Study power for adjusted models.**
(DOCX)

## Author Contributions

**Conceptualization:** Fernanda de Souza Nogueira Sardinha Mendes, Paula Simplício da Silva, Gilberto Marcelo Sperandio da Silva, Andrea Silvestre de Sousa, Roberto Magalhães Saraiva, Alejandro Marcel Hasslocher-Moreno, Pedro Emmanuel Alvarenga Americano do Brasil, Mauro Felippe Felix Mediano.

**Data curation:** Fernanda de Souza Nogueira Sardinha Mendes, Paula Simplício da Silva, Gilberto Marcelo Sperandio da Silva, Andrea Silvestre de Sousa, Roberto Magalhães Saraiva, Alejandro Marcel Hasslocher-Moreno, Pedro Emmanuel Alvarenga Americano do Brasil, Mauro Felippe Felix Mediano.

**Formal analysis:** Marcelo Carvalho Vieira, Marcel de Souza Borges Quintana, Mauro Felippe Felix Mediano.

**Funding acquisition:** Mauro Felippe Felix Mediano.

**Investigation:** Fernanda de Souza Nogueira Sardinha Mendes, Paula Simplício da Silva, Gilberto Marcelo Sperandio da Silva, Andrea Silvestre de Sousa, Roberto Magalhães Saraiva, Alejandro Marcel Hasslocher-Moreno, Pedro Emmanuel Alvarenga Americano do Brasil, Mauro Felippe Felix Mediano.

**Methodology:** Fernanda de Souza Nogueira Sardinha Mendes, Mauro Felippe Felix Mediano.

**Project administration:** Fernanda de Souza Nogueira Sardinha Mendes, Mauro Felippe Felix Mediano.

**Resources:** Mauro Felippe Felix Mediano.

**Software:** Mauro Felippe Felix Mediano.

**Supervision:** Fernanda de Souza Nogueira Sardinha Mendes, Mauro Felippe Felix Mediano.

**Validation:** Mauro Felippe Felix Mediano.

**Visualization:** Marcelo Carvalho Vieira, Mauro Felippe Felix Mediano.

**Writing – original draft:** Marcelo Carvalho Vieira, Mauro Felippe Felix Mediano.

**Writing – review & editing:** Marcelo Carvalho Vieira, Fernanda de Souza Nogueira Sardinha Mendes, Paula Simplício da Silva, Gilberto Marcelo Sperandio da Silva, Flavia Mazzoli-Rocha, Andrea Silvestre de Sousa, Roberto Magalhães Saraiva, Marcel de Souza Borges Quintana, Henrique Silveira Costa, Vitor Barreto Paravidino, Luiz Fernando Rodrigues, Junior, Alejandro Marcel Hasslocher-Moreno, Pedro Emmanuel Alvarenga Americano do Brasil, Mauro Felippe Felix Mediano.

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
