## [Decision Letter · Decision Letter 0]

26 Aug 2022

PONE-D-22-07420

The association between variables of cardiopulmonary exercise test and quality of life in patients with chronic Chagas cardiomyopathy (Insights from the PEACH STUDY)

PLOS ONE

Dear Dr. Vieira,

Thank you for submitting your manuscript to PLOS ONE. After careful consideration, we feel that it has merit but does not fully meet PLOS ONE’s publication criteria as it currently stands. Therefore, we invite you to submit a revised version of the manuscript that addresses the points raised during the review process.

Dear Dr. Marcelo Carvalho Vieira

Thank you for submitting your manuscript to *PLOS ONE*. After careful consideration, we feel that it has merit, but is not suitable for publication as it currently stands. Therefore, my decision is "Major Revision.” We invite you to submit a revised version of the manuscript that addresses the points below: 

I am returning your manuscript with four reviews. The reviewers came to different conclusions about the paper, as you will see. After reading the reviews and looking at the manuscript, I am afraid that I have to concur with the more critical review. I am sorry I cannot be more positive at the moment, but as I have noted, all is not lost. It requires a lot of work and a major revision that I believe that you may need more time to work on the manuscript for a resubmission if you so wish to do so. 

Note that it will have to go through the second round of review. 

Please pay attention to the following reviewer suggestions and give them due consideration.

We encourage you to submit your revision within sixty days of the date of this decision. When your files are ready, please submit your revision by logging on to http://pone.edmgr.com/ and following the Submissions Needing Revision link. Do not submit a revised manuscript as a new submission. 

Please also include a rebuttal letter that responds to each point brought up by the Academic Editor and reviewer(s). This letter should be uploaded as a Response to Reviewers file. 

In addition, please provide a marked-up copy of the changes made from the previous article file as a Manuscript with Tracked Changes file. This can be done using 'track changes' in programs such as MS Word and/or highlighting any changes in the new document.

If you choose not to submit a revision, please notify us. 

We look forward to receiving your revised manuscript.

Kind regards,

Gerson Cipriano Jr., PT, MsC, Ph.D.

Academic Editor

PLOS ONE

https://journals.plos.org/plosone/s/file?id=ba62/PLOSOne_formatting_sample_title_authors_affiliations.pdf".

Reviewers' comments:

Reviewer's Responses to Questions

**Comments to the Author**

1. Is the manuscript technically sound, and do the data support the conclusions?

Reviewer #1: Yes

Reviewer #2: Yes

Reviewer #3: Yes

Reviewer #4: Yes

2. Has the statistical analysis been performed appropriately and rigorously? 

Reviewer #1: Yes

Reviewer #2: Yes

Reviewer #3: Yes

Reviewer #4: Yes

3. Have the authors made all data underlying the findings in their manuscript fully available?

Reviewer #1: Yes

Reviewer #2: Yes

Reviewer #3: No

Reviewer #4: Yes

4. Is the manuscript presented in an intelligible fashion and written in standard English?

Reviewer #1: Yes

Reviewer #2: Yes

Reviewer #3: Yes

Reviewer #4: Yes

5. Review Comments to the Author

Reviewer #1: The present is an interesting paper aiming to evalaute and standardize evaluation of patients with Chagas Disease

Some issues remain to be address

1)in the abstract some numerical data regarding HR/OR should be added (at least those more relevant)

2) In the introduction it should be better specified the importance of a "standardization"of QoL.

3) was the present analysis pre-specified or not?

4) Usually evaluation with time consuming questionare is challenging. Who asked the questions to the patients? DO authors recorded a % of complete or not answers?

5) regarding CPET did authors recorded % of medications?

6) Did authors performed a MRI to these patients or not?

7) due to reduced sample size, did authors checked for normality?

8) the rest of the statistical anakysis is correct. Usually i do not like when authors correct for "variables knowm in literature", anyway in this case it is accpetable due to reduced sample size. Please comment

Reviewer #2: The association between variables of cardiopulmonary exercise test and quality of life in patients with chronic Chagas cardiomyopathy (Insights from the PEACH STUDY) - PONE-D-22-07420

1. I suggest the authors put the of the CEP number 2908991, once the CAAE is just a number generated to identify the research project that enters for ethical consideration in the CEP.

2. Consider to mention that this is an arm of a clinical study, since in the clinical trails the main objectives were to evaluate interventions such as exercise, nutritional counseling and pharmaceuticals and not the association of cardiopulmonary variables and quality of life.

3. Was an a priori sample calculation performed? If yes, what was the outcome variable used? If not, I suggest calculating the test power.

I understand that as this is an arm of a clinical study, the sample size calculation is not presented and/or applied to the study in question.

Reviewer #3: In the present study the authors analize the correlation between quality of life and exercise performance in patients with Chagas cardiomyopathy. The study is confirmatory of this assosiation in heart failure patients, albeit, as far as I know, not previously evaluated in patients with Chagas cardiomyopathy.

The study is well presented, but I suggest the authors to add among the variables VO2 reported as a percent of predicted.

The same for the VE/VCO2 relationship slope.

The discussion is by far too long and I believe that it should be shortened. Specifically all the parts about autonomic disfunction is mainly speculative and should be deleted.

Reviewer #4: The study entitled "the association between variables of cardiopulmonary exercise test and quality of life in patients with chronic Chagas cardiomyopathy (Insights from the PEACH STUDY)" is well written and presents an interesting topic of a neglected disease with reflections on other cardiomyopathies. Some aspects need further clarification.

1. Can you provide the interval between CPET and Echocardiography. This is relevant to ascertain that it may reflect precisely its relation to CPET.

2. On page 8 line 190, you stated that a HRR<12bpm is probably related to autonomic dysfunction. I suggest considering and presenting data corroborating that. A non-trained individual may present a reduced HRR so looking at a VO2 may help exclude that possibility.

3. Was the CPET performed using Beta-blockers? If not, please insert that information. If they were performed using the medications I suggest a paragraph about the possible impact on the results, since nearly all sample was taking them. IT may be a study limitation.

4. Age, gender, and LVEF were your potential confounders, but the literature also presents overweight and obesity as potential confounders (PLoS One. 2021; 16(8): e0255724.). Why you did not consider them? Nearly half of your sample has overweight or obese. Please comment on that.

5. Although the discussion is comprehensive, I suggest a paragraph on study limitations, discussing beta blockers and overweight impact and the absence of a direct evaluation of the autonomic function, as well as the small sample evaluated.

6. PLOS authors have the option to publish the peer review history of their article (what does this mean?). If published, this will include your full peer review and any attached files.

Reviewer #1: **Yes: **Fabrizio D'Ascenzo

Reviewer #2: **Yes: **Daniela Bassi Dibai

Reviewer #3: **Yes: **oiergiuseppe agostoni

Reviewer #4: No

---

## [Author Response · Author response to Decision Letter 0]

1 Nov 2022

https://journals.plos.org/plosone/s/file?id=ba62/PLOSOne_formatting_sample_title_authors_affiliations.pdf".

Response: Thanks for the notice. We have reviewed the manuscript and ensured that it meets all PLOS ONE style requirements.

2) In your Data Availability statement, you have not specified where the minimal data set underlying the results described in your manuscript can be found. PLOS defines a study's minimal data set as the underlying data used to reach the conclusions drawn in the manuscript and any additional data required to replicate the reported study findings in their entirety. All PLOS journals require that the minimal data set be made fully available. For more information about our data policy, please see http://journals.plos.org/plosone/s/data-availability.

Response: We apologize for the misunderstanding. The dataset is now available at osf.io/gkavb. Thanks for updating the respective statement.

3) We note that you have indicated that data from this study are available upon request. PLOS only allows data to be available upon request if there are legal or ethical restrictions on sharing data publicly. For more information on unacceptable data access restrictions, please see http://journals.plos.org/plosone/s/data-availability#loc-unacceptable-data-access-restrictions.

Response: We apologize for the misunderstanding. The dataset is now available at osf.io/gkavb. Thanks for updating the respective statement. 

Reviewer 1: The present is an interesting paper aiming to evaluate and standardize evaluation of patients with Chagas Disease. Some issues remain to be address.

1) In the abstract some numerical data regarding HR/OR should be added (at least those more relevant)

Response: We thank the reviewer for the suggestion. We added the β coefficients for all the significant variables in the abstract.

2) In the introduction it should be better specified the importance of a "standardization" of QoL.

Response: We appreciated the reviewer suggestion. We added a sentence highlighting the importance of HRQoL standardization.

3) Was the present analysis pre-specified or not?

Response: This is a secondary analysis from the PEACH clinical trial using the baseline data. This information was included in the manuscript.

4) Usually evaluation with time consuming questionnaire is challenging. Who asked the questions to the patients? Do authors recorded a % of complete or not answers?

Response: The same investigator was responsible for the application of the questionnaires in all patients, which took approximately 20 to 30 minutes. No patient failed to answer the questionnaire. This information was included in the manuscript.

5) Regarding CPET did authors recorded % of medications?

Response: The CPET was performed with patients taking all their medications. This information was included in the manuscript. The medications in use by the patients at the time of CPET are described in the Table 1.

6) Did authors perform a MRI to these patients or not?

Response: MRI is not a standard exam for evaluation of Chagas cardiomyopathy and was not performed in any patient. 

7) Due to reduced sample size, did authors checked for normality? 

Response: We use generalized linear models with gamma distribution and log-link function that accounts for skewed and heteroscedastic residuals distribution. This information is included in the manuscript. 

8) The rest of the statistical analysis is correct. Usually I do not like when authors correct for "variables known in literature", anyway in this case it is acceptable due to reduced sample size. Please comment

Response: Many thanks for the comment. Age, sex, and left ventricle ejection fraction are the variables most related to prognosis in Chagas cardiomyopathy and, because of this, they were considered in our model as potential confounders.

 

Reviewer 2

1) I suggest the authors put the of the CEP number 2908991, once the CAAE is just a number generated to identify the research project that enters for ethical consideration in the CEP.

Response: We appreciated the reviewer suggestion. We added the report number after the CAAE.

2) Consider to mention that this is an arm of a clinical study, since in the clinical trails the main objectives were to evaluate interventions such as exercise, nutritional counseling and pharmaceuticals and not the association of cardiopulmonary variables and quality of life.

Response: We thank the reviewer for this comment. We rephrased this sentence in the methods section in order to improve clarity. 

3) Was an a priori sample calculation performed? If yes, what was the outcome variable used? If not, I suggest calculating the test power. I understand that as this is an arm of a clinical study, the sample size calculation is not presented and/or applied to the study in question.

Response: The sample size was calculated only for the clinical trial to detect a clinically significant difference between groups in peak VO2. We calculated the study power for the analysis performed in this secondary analysis, which ranged from 5% to 97%. Please see supplementary table. This information was included in the manuscript. 

 

Reviewer 3: In the present study the authors analyze the correlation between quality of life and exercise performance in patients with Chagas cardiomyopathy. The study is confirmatory of this association in heart failure patients, albeit, as far as I know, not previously evaluated in patients with Chagas cardiomyopathy.

1) The study is well presented, but I suggest the authors to add among the variables VO2 reported as a percent of predicted.

Response: We thank the reviewer for the suggestion. We include the variable in the analysis as suggested. The results are described in the text, in Tables 1 and 3 and in Figure 1. 

2) The same for the VE/VCO2 relationship slope.

Response: Mean and standard deviation of the VE/VCO2 slope of participants included in the study were presented in the Table 1. The association between VE/VCO2 slope and QoL scales were presented in the Tables 3 and 4.

3) The discussion is by far too long and I believe that it should be shortened. Specifically all the parts about autonomic disfunction is mainly speculative and should be deleted.

Response: We thank the reviewer for this comment. However, although we do agree that the explanations for our findings on autonomic modulation are hypothetical, they are supported by some references cited in the text. We believe that these mechanisms should be presented, even to stimulate further research on the subject.

 

Reviewer 4: The study entitled "The association between variables of cardiopulmonary exercise test and quality of life in patients with chronic Chagas cardiomyopathy (Insights from the PEACH STUDY)" is well written and presents an interesting topic of a neglected disease with reflections on other cardiomyopathies. Some aspects need further clarification.

1) Can you provide the interval between CPET and Echocardiography? This is relevant to ascertain that it may reflect precisely its relation to CPET.

Response: Thank you for the question. CPET and echo were performed within one-week range. This information was included in the manuscript.

2) On page 8 line 190, you stated that a HRR<12bpm is probably related to autonomic dysfunction. I suggest considering and presenting data corroborating that. A non-trained individual may present a reduced HRR so looking at a VO2 may help exclude that possibility.

Response: We thank the reviewer for this comment. It has been demonstrated that a blunted heart rate recovery after exercise testing is a predictor of mortality, both in individuals without cardiovascular disease (Cole et al., 1999) and in individuals with heart failure (Bilsel et al., 2006; Arena et al., 2010). Moreover, Imai et al. (1994) demonstrated that heart rate recovery after exercise testing is blunted in heart failure patients even when compared to age-matched normal control individuals, which presented a statistically significant lower VO2. All the above cited references are included in our paper. We made changes to the text to clarify this point.

3) Was the CPET performed using Beta-blockers? If not, please insert that information. If they were performed using the medications I suggest a paragraph about the possible impact on the results, since nearly all sample was taking them. It may be a study limitation.

Response: Yes, the CPET was performed with patients taking all their medications, including beta-blockers (93.3%) when applicable. We have included a sentence on study limitations on the impact of beta-blocker use on CPET responses.

4) Age, gender, and LVEF were your potential confounders, but the literature also presents overweight and obesity as potential confounders (PLoS One. 2021; 16(8): e0255724.). Why did you not consider them? Nearly half of your sample has overweight or obese. Please comment on that.

Response: We thank the reviewer for the suggestion. Although overweight/obesity has an influence on the CPET response, we chose to include as potential confounders those variables whose impact on the morbidity and mortality of patients with CCC has already been reported in the literature, such as age, gender, and LVEF. In addition, the small sample size prevented us from including more confounding variables in the statistical model.

5) Although the discussion is comprehensive, I suggest a paragraph on study limitations, discussing beta blockers and overweight impact and the absence of a direct evaluation of the autonomic function, as well as the small sample evaluated.

Response: Thank you very much for the suggestion. We have expanded the section on study limitations to include the reviewer's suggestions.

---

## [Editor Report · Decision Letter 1]

18 Nov 2022

PONE-D-22-07420R1The association between variables of cardiopulmonary exercise test and quality of life in patients with chronic Chagas cardiomyopathy (Insights from the PEACH STUDY)PLOS ONE

Dear Dr. Vieira,

Thank you for submitting your manuscript to PLOS ONE. After careful consideration, we feel that it has merit but does not fully meet PLOS ONE’s publication criteria as it currently stands. Therefore, we invite you to submit a revised version of the manuscript that addresses the points raised during the review process.

Thank you for submitting your paper to PlosOne. We have now completed our review.This original research aimed to evaluate the association between functional capacity (quantified by cardiopulmonary exercise test [CPET]) and QoL in individuals with CCC. 

Your manuscript was thoughtfully written and could potentially make an impactful contribution to the scientific literature; however, we want to ask you to complete a review of your figure 1, which must be extensively reviewed according to Plos One Figure preparation review and possible professionally redesign for better illustrate your study statement.

We look forward to receiving your revised manuscript.

Kind regards,

Gerson Cipriano Jr., PT, MsC, Ph.D.

Academic Editor

PLOS ONE
---

## [Author Response · Author response to Decision Letter 1]

22 Nov 2022

To Editorial Board of PLOS ONE

Dear Editor-in-Chief, 

Manuscript ID: PONE-D-22-07420

Title: The association between variables of cardiopulmonary exercise test and quality of life in patients with chronic Chagas cardiomyopathy (Insights from the PEACH STUDY)

Dear editor,

Thank you very much for giving us the opportunity to review our Figure 1. The figure has been revised according to PLOS ONE’s “Figure preparation instructions” and the figure file has been uploaded to the PACE digital diagnostic tool to meet journal requirements. We hope the paper is now suitable for publication at PLOS ONE. Please, feel free to contact me in case of any addition doubt.

Sincerely yours,

Marcelo Carvalho Vieira, BPhEd., MSc., PhD

Evandro Chagas National Institute of Infectious Disease, Oswaldo Cruz Foundation, Rio de Janeiro, RJ, Brazil

---

## [Editor Report · Decision Letter 2]

1 Dec 2022

The association between variables of cardiopulmonary exercise test and quality of life in patients with chronic Chagas cardiomyopathy (Insights from the PEACH STUDY)

PONE-D-22-07420R2

Dear Dr. Vieira,

We’re pleased to inform you that your manuscript has been judged scientifically suitable for publication and will be formally accepted for publication once it meets all outstanding technical requirements.

Kind regards,

Gerson Cipriano Jr., PT, MsC, Ph.D.

Academic Editor

PLOS ONE
---

## [Editor Report · Acceptance letter]

6 Dec 2022

PONE-D-22-07420R2 

The association between variables of cardiopulmonary exercise test and quality of life in patients with chronic Chagas cardiomyopathy (Insights from the PEACH STUDY) 

Dear Dr. Vieira:

I'm pleased to inform you that your manuscript has been deemed suitable for publication in PLOS ONE. Congratulations! Your manuscript is now with our production department. 

Kind regards, 

on behalf of

Professor Gerson Cipriano Jr. 

Academic Editor

PLOS ONE